# An Efficient LLM Alignment Framework for Automated Radiology Impression Generation

## Abstract

Large language models (LLMs) are typically specialized for domain tasks through supervised fine-tuning, which optimizes LLMs for likelihood-based objectives. While supervised fine-tuning enables LLMs to generate text that conforms to the language style of a specific domain, such as radiology, it often falls short in enhancing the model's ability to perform detailed diagnostic reasoning that tailors to the need of individual patients. In this paper, we explore the use of reinforcement learning to better align LLMs with the intricate requirements of radiological practice. By framing the report generation process as sequential decision-making stages, we present Radiology-Guided Reinforcement Optimization (RGRO), a tailored policy optimization framework designed specifically for medical language tasks. RGRO moves beyond conventional likelihood-based training by directly optimizing for radiology-specific objectives, including consistency with radiology findings and adherence to established professional guidelines. Our empirical evaluations demonstrate that RGRO significantly enhances the diagnostic precision and clinical utility of radiology reports generated by LLMs, outperforming supervised fine-tuning methods and state-of-the-art models. Furthermore, RGRO enables the seamless integration of expert radiologist feedback and external diagnostic tools, all without the need for large-scale annotated datasets.

## 1 Introduction

The advent of large language models (LLMs) marks a major milestone in the field of natural language processing, and has led to unprecedented advancements in various fields (Elyoseph et al., 2024; Dagdelen et al., 2024; Wagner & Ertl-Wagner, 2023) such as text generation, text comprehension, and interactive dialogue. To adapt to target use cases, models are typically fine-tuned on large corpora using supervised learning techniques that optimize likelihood-based objectives (Min et al., 2023). This fine-tuning process enables them to excel at generating coherent and contextually relevant text, aligning with the patterns observed in the training data. However, while fine-tuning enhances language contextualization and stylistic adherence, it does not address the deeper requirements of specialized domains, where reasoning and insights are paramount (Bhayana, 2024; Tang et al., 2023). Specifically, radiologists generate reports by synthesizing both positive and negative findings from medical images before forming a diagnostic impression (Omiye et al., 2024). This process demands not only language fluency but also meticulous diagnostic reasoning, adherence to clinical standards, and relevance to individual patient cases. Given the high stakes involved in medical diagnoses, precision and reliability in generated reports are crucial. These challenges call for a paradigm shift in how LLMs are trained and applied in specialized fields.

Conventional methods for adapting LLMs to radiology rely on prompt engineering and supervised fine-tuning (Yan et al., 2024; Ma et al., 2024; Jiang et al., 2023; Luo et al., 2022). While these approaches can infuse some domain-specific features into the model, it is heavily reliant on the availability of large-scale, high-quality annotated data—a scarce resource in the medical field (Bhayana, 2024; Liu et al., 2023a) due to confidentiality concerns, regulatory restrictions, and the considerable effort required for expert annotations. Moreover, traditional likelihood-based training objectives, which aim to maximize the probability of observed text sequences, may not align with the nuanced demands of radiological practice (Lecler et al., 2023). This misalignment can result in outputs that, although linguistically coherent, lack clinical efficacy or fail to adhere to professional guidelines (Zhong et al., 2023). This gap underscores a critical challenge in deploying LLMs for decision

reasoning process and highlights the need for the alignment strategies tailored to domain-specific requirements.

In this paper, we present radiology-guided reinforcement optimization (RGRO), a novel policy optimization framework (Buffet et al., 2020) designed explicitly for radiology. By modeling the radiology report generation process as a continuous multi-stage decision phase, we first leverage instruction fine-tuning to adapt LLMs to comprehend and synthesize detailed findings into coherent impressions (Nie et al., 2018). This initial phase conditions the model on the unique characteristics of radiological language, training it to distill intricate imaging observations into meaningful summaries. Subsequently, RGRO employs reinforcement learning guided by enhancement feedback leveraging generative language models to iteratively align the model's outputs with expert-driven benchmarks based on explicit reward signals and directly optimizes for radiology-specific objectives. This approach reduces reliance on the typically scarce and costly large-scale annotated datasets in the medical domain. Distinctly differing from traditional likelihood-based training schemes, RGRO focuses on enhancing fine-grained reasoning (Cosentino & Shekkizhar, 2024) performance within a specific domain. Rather than relying solely on likelihood maximization, we directly optimize for domain-specific goals. This perspective allows the model to learn strategies that are not only generally plausible but are meticulously tailored to the radiological context.

Our contributions include:

- We develop RGRO, a reinforcement learning-based policy optimization framework specifically tailored for medical language tasks, which transforms the report generation process into a sequential decision-making problem.
- By formulating custom reward functions, RGRO directly optimizes critical domain-specific objectives, such as clinical accuracy, diagnostic reasoning, and adherence to established reporting standards.
- Our approach enables the incorporation of pre-existing expert radiologist insights and external diagnostic information, which reduces the dependency on large-scale annotated datasets and addresses data scarcity challenges.

Through comprehensive experiments, we demonstrate that RGRO significantly improves the diagnostic precision and clinical utility of generated radiology reports, outperforming the state-of-the-art models. While tailored for radiology, our framework lays the groundwork for applying reinforcement learning to other specialized domains requiring precise and context-specific tasks.

## 2 RELATED WORK

**The general capabilities of large language models.** Large language models have demonstrated impressive capabilities in natural language understanding and generation. They can perform various tasks such as text summarization, translation, question answering, sentiment analysis, and beyond (Zhou et al., 2023). The expressive transformer architecture (Vaswani, 2017) enables LLMs to capture complex relationships within data. Furthermore, fine-tuning techniques enable LLMs to adapt to specific tasks or domains, which enhances their performance and applicability across different fields.

**Large language models for radiology report generation.** In the medical domain, large language models (Vaswani, 2017) have shown significant potential in generating radiology reports (Ma et al., 2024; Bhayana, 2024; Jiang et al., 2023; Liu et al., 2023b). By understanding the context and structure of radiology findings (Yuan et al., 2019; Chen et al., 2020), these models can assist in creating accurate and coherent reports. The ability of LLMs to synthesize information from various sources allows for the generation of reports that adhere to clinical standards while ensuring clarity and precision (Liu et al., 2019).

**Reasoning capabilities of large language models.** Large language models exhibit notable reasoning capabilities that enable them to process and analyze complex information (Kojima et al., 2022). They can perform logical reasoning, make inferences, and understand nuanced prompts, which is essential for tasks requiring deep comprehension. These reasoning abilities arise from training on diverse datasets, which allows LLMs to understand context, relationships, and implications (Li et al., 2024). However, since LLMs are autoregressive models that generate text one token at a time based

on previously generated tokens, they often face challenges in maintaining consistent and coherent reasoning over long sequences. This autoregressive nature (Dalal et al., 2019) can lead to difficulties in handling complex logical structures and maintaining accuracy across multiple steps of reasoning, limiting their performance in tasks requiring deep, multi-step inference (Kalyanpur et al., 2024).

**Applying reinforcement learning to align with physicians.** Reinforcement learning (RL) has become crucial for enhancing decision-making in large language models, refining model behavior based on user feedback (Kaufmann et al., 2023; Ouyang et al., 2022). Previously, Proximal Policy Optimization (PPO) (Schulman et al., 2017), a reinforcement learning algorithm, demonstrated how policies could be efficiently updated using a surrogate objective, ensuring stability while allowing multiple updates. More recently, Direct Preference Optimization (DPO) (Rafailov et al., 2023) has been introduced to align models with human preferences while retaining their domain knowledge, ensuring that fine-tuning does not compromise the expertise gained during pretraining. Applying DPO to radiology impression generation represents a novel approach, offering a promising way to adapt LLMs to meet the specific requirements of medical professionals.

## 3 PRELIMINARIES

In this section, we briefly review the key concepts and techniques from reinforcement learning that serve as the foundation for our proposed framework.

### 3.1 MARKOV DECISION PROCESS (MDP)

A reinforcement learning problem is typically modeled as a Markov Decision Process (MDP) (Black et al., 2023). An MDP is defined by a tuple $(\mathcal{S}, \mathcal{A}, P, r, \gamma)$ where:

- $\mathcal{S}$ is the state space, representing all possible states of the environment.
- $\mathcal{A}$ is the action space, which includes all possible actions the agent can take.
- $P(s'|s, a)$ is the transition probability, representing the probability of reaching state $s'$ from state $s$ after taking action $a$.
- $r : \mathcal{S} \times \mathcal{A} \rightarrow \mathbb{R}$ is the reward function, which provides feedback to the agent based on the action taken.
- $\gamma \in [0, 1]$ is the discount factor, which trades off the importance of immediate and future rewards.

The objective in reinforcement learning (Flageat et al., 2023) is to learn a policy $\pi(a|s)$ that maximizes the expected cumulative reward (return), defined as:

$$\mathbb{E}\left[\sum_{t=0}^{\infty} \gamma^t r(s_t, a_t)\right]$$

where $s_t$ and $a_t$ are the state and action at time step $t$.

### 3.2 POLICY OPTIMIZATION

Policy optimization methods aim to directly optimize the policy $\pi_\theta$ parameterized by $\theta$. One common approach is to use policy gradient methods (Black et al., 2023), which optimize the expected return by estimating the gradient of the objective with respect to the policy parameters:

$$\nabla_\theta J(\pi_\theta) = \mathbb{E}_{s \sim \rho^\pi, a \sim \pi_\theta} \left[\nabla_\theta \log \pi_\theta(a|s) Q^\pi(s, a)\right]$$

where $Q^\pi(s, a)$ is the action-value function under the policy $\pi_\theta$ and $\rho^\pi(s)$ is the stationary distribution of states.

### 3.3 PROXIMAL POLICY OPTIMIZATION (PPO)

Proximal Policy Optimization (PPO) (Schulman et al., 2017; Yu et al., 2022) is a policy gradient method designed to maintain a balance between exploration and exploitation. It prevents large

updates to the policy by using a clipped objective function:

$$L^{\text{PPO}}(\theta) = \mathbb{E}_t \left[ \min \left( r_t(\theta)\hat{A}_t, \text{clip}(r_t(\theta), 1 - \epsilon, 1 + \epsilon)\hat{A}_t \right) \right]$$

where $r_t(\theta)$ is the probability ratio between the new and old policy, and $\hat{A}_t$ is the advantage estimate. The clipping mechanism ensures that the policy update is conservative, improving the stability of training.

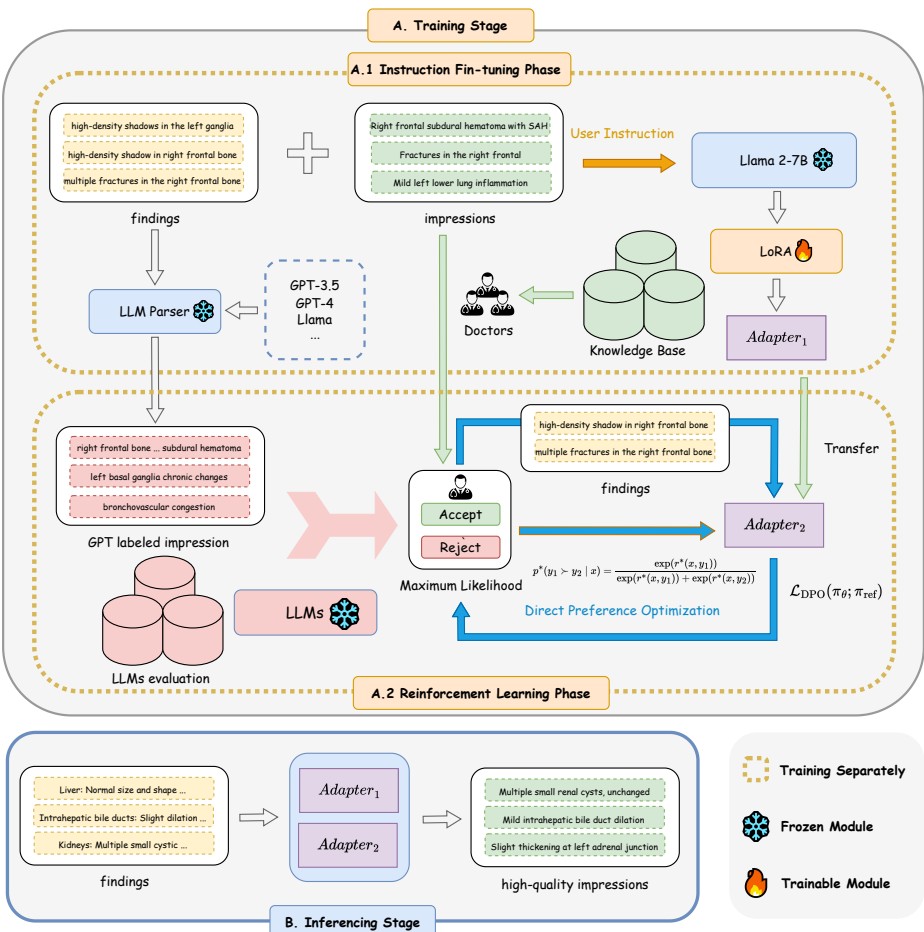

**Figure** 1: **Radiology-Guided Reinforcement Optimization.** We propose a reinforcement learning framework, RGRO, to optimize large language models on clinical objectives of radiology reports. Leveraging this framework, we develop a finding-impression alignment structure with radiologists knowledge. Each row shows the detailed process of each stage of the different training phases, such as data evaluation, clinical processing and model optimization.

## 4 RADIOLOGY-GUIDED REINFORCEMENT OPTIMIZATION

In this section, we detail the two phases used to develop our framework: the *Instruction Fine-tuning* phase and the *Reinforcement Learning* phase, as illustrated in Figure 1. Our objective is to generate radiology impressions that are more closely aligned with doctors' expectations, ensuring they are both coherent and clinically accurate based on the *findings* in radiology reports.

### 4.1 INSTRUCTION FINE-TUNING WITH LoRA

In the instruction fine-tuning phase, we aim to teach the model to transform the findings in radiology reports into structured and accurate impression sections. By integrating the Low-Rank Adaptation

(LoRA) technique (Hu et al., 2021), this phase becomes crucial for establishing a robust foundation in medical impression generation.

We utilize a large-scale radiology dataset where each sample consists of a *finding-impression* pair, denoted as $\mathcal{D}_{FT} = \{(x_i, y_i)\}_{i=1}^{N}$, where $x_i$ represents the findings (e.g., detailed descriptions of radiological observations), and $y_i$ represents the corresponding impression (e.g., summarized diagnostic conclusions or recommendations).

LoRA allows us to adapt the pre-trained model efficiently by introducing trainable low-rank matrices into each layer of the Transformer architecture, significantly reducing the number of trainable parameters required for fine-tuning. Specifically, given a weight matrix $W \in \mathbb{R}^{d \times k}$ in a pre-trained model, LoRA approximates the adaptation of $W$ by representing it as:

$$W' = W + AB$$

where $A \in \mathbb{R}^{d \times r}$ and $B \in \mathbb{R}^{r \times k}$ are low-rank matrices with rank $r \ll \min(d, k)$. During fine-tuning, only $A$ and $B$ are updated, while the original weights $W$ remain fixed, allowing for a more memory-efficient adaptation process.

The goal during this phase is to optimize the model's parameters $\theta$ (in this case, the parameters of matrices $A$ and $B$ introduced by LoRA) to generate clinically accurate impressions based on the given findings. To achieve this, we minimize the cross-entropy loss:

$$\mathcal{L}_{FT}(\theta) = -\frac{1}{N} \sum_{i=1}^{N} \log \pi_\theta(y_i|x_i),$$

where $\pi_\theta(y_i|x_i)$ represents the probability that the model assigns to the correct impression $y_i$ given the findings $x_i$. This loss function encourages the model to produce impressions that closely match those written by expert radiologists in the dataset. By leveraging the LoRA method, the model efficiently learns to interpret findings and generate concise and informative impressions, establishing a strong baseline for subsequent reinforcement learning optimization.

## 4.2 LLM PARSER AND PREFERENCE OPTIMIZATION FOR HUMAN FEEDBACK

In this study, we utilized LLM Parser (OpenAI, 2023) as a substitute for human evaluators to streamline the feedback process. Instead of relying on human annotators to provide preference feedback on the model's outputs, we leveraged LLM Parser to select the preferred output between two generated responses. Specifically, for each pair of outputs from the model, LLM Parser was tasked with determining which output better aligned with the desired criteria, such as accuracy, relevance, and clarity. This approach allowed for a more efficient and consistent evaluation process, reducing the reliance on human involvement while maintaining a high standard of feedback quality.

After that, our approach leverages a modified version of Direct Preference Optimization (DPO) (Rafailov et al., 2023) to align the model's behavior with expert radiologists' preferences in generating clinically relevant impressions from findings. Instead of strictly following the original DPO methodology, we directly use impressions written by radiologists during their routine work as the accepted responses, while the samples unpreferred by LLM Parser serve as the rejected responses.

In this framework, we construct preference data as tuples $(f_i, imp_a, imp_r)$, where $f_i$ represents the detailed radiology findings, and $imp_a$ and $imp_r$ are two impressions generated by the model. The LLM Parser (acting as the evaluator) is used to decide which impression to "accept" ($imp_a$) and which to "reject" ($imp_r$), based on alignment with expert radiologist annotations. By comparing high-quality, expert-aligned impressions with those that are deemed suboptimal by the LLM Parser, our approach leverages domain-specific expertise to guide the model's learning process. By employing the LLM Parser as a judge instead of relying only on traditional preference data, this approach brings a fresh variation to the standard DPO framework. It allows the model to engage with a wide range of preferred and unpreferred outputs. This contrast between expert-level impressions and less optimal ones provides the model with a richer learning signal, which helps it to capture the subtleties that distinguish truly expert-level impressions from samples that are superficially coherent. The loss function is defined as:

$$\mathcal{L}_{DPO}(\pi_\theta; \pi_{ref}) = -\mathbb{E}_{(f_i, imp_a, imp_r) \sim \mathcal{D}} \left[ \log \sigma \left( \beta \log \frac{\pi_\theta(imp_a|f_i)}{\pi_{ref}(imp_a|f_i)} - \beta \log \frac{\pi_\theta(imp_r|f_i)}{\pi_{ref}(imp_r|f_i)} \right) \right],$$

where $\pi_\theta$ represents the policy we aim to optimize, $\pi_{ref}$ is the reference policy obtained from the instruction fine-tuning phase, and $\beta$ is a scaling factor that controls the sensitivity of the loss function to preference differences. This loss function guides the model in distinguishing between clinically relevant and less relevant impressions, allowing it to refine its outputs towards meeting the high standards expected in radiological reporting, ultimately aligning more closely with the nuanced preferences of medical professionals.

To ensure the model generates high-quality impressions, the reward function $r(f_i, imp)$, which measures how well an impression aligns with the accepted dataset labels:

$$r(f_i, imp) = \beta \log \frac{\pi_\theta(imp|f_i)}{\pi_{ref}(imp|f_i)} + \beta \log Z(f_i),$$

where $Z(f_i)$ is a normalization term. This reward function evaluates the alignment of generated impressions with the expert-labeled data over the LLM Parser generated alternatives.

The preference probability, indicating the likelihood that the accepted impression $imp_a$ is preferred over the rejected impression $imp_r$, is calculated as:

$$p^*(imp_a \succ imp_r \mid f_i) = \sigma \left( r(f_i, imp_a) - r(f_i, imp_r) \right),$$

or equivalently:

$$p^*(imp_a \succ imp_r \mid f_i) = \frac{1}{1 + \exp \left( \beta \log \frac{\pi_\theta(imp_r|f_i)}{\pi_{ref}(imp_r|f_i)} - \beta \log \frac{\pi_\theta(imp_a|f_i)}{\pi_{ref}(imp_a|f_i)} \right)}.$$

This framework ensures that the model learns to prefer impressions aligned with radiologist-labeled data over less accurate LLM Parase-generated impressions. By optimizing the model in this way, this process ensures that the generated outputs not only meet clinical and diagnostic standards but also align more closely with the expectations of medical professionals.

## 5 EXPERIMENTAL EVALUATIONS

We designed our experiments to systematically evaluate the performance of our model in generating radiology impressions from findings. The experiments are divided into multiple phases, each addressing different aspects of model training and evaluation, including ablation studies, comparative experiments, multi-center generalization, and visualizations. For the experimental setup, we utilized 12 A100 GPUs to train and 8 A5000 GPUs to conduct our tests efficiently.

### 5.1 EVALUATION METRICS: ROUGE AND BERTSCORE

To evaluate the performance of our model in generating clinically relevant and accurate radiology impressions, we employed two widely used metrics: ROUGE and BERTScore.

**ROUGE (Lin, 2004) (Recall-Oriented Understudy for Gisting Evaluation):** ROUGE is a set of metrics for evaluating text generation by comparing the overlap between the generated text and reference text. ROUGE-N measures n-gram overlap, such as Rouge-1 (R1) and Rouge-2 (R2), while ROUGE-L (RL) focuses on the longest common subsequence. Higher ROUGE scores indicate greater similarity to the reference, making it useful for assessing alignment with expert annotations.

**BERTScore:** BERTScore (Zhang et al., 2020) uses pre-trained BERT embeddings to evaluate the semantic similarity between generated and reference texts. Unlike ROUGE, which focuses on token overlap, BERTScore captures deeper contextual meaning by computing cosine similarity between

token embeddings, making it ideal for assessing clinically relevant impressions. By combining ROUGE and BERTScore, we evaluate both lexical similarity and semantic alignment with expert-generated impressions.

## 5.2 PHASE 1: ABLATION AND COMPARATIVE STUDY

**Experimental Design:** We conducted an ablation study to assess different training methodologies for generating radiology impressions from findings. Using LLAMA2-7B (Touvron et al., 2023) as the foundational model, pre-trained on a radiology dataset, we fine-tuned it on two prominent datasets: *MIMIC* (Johnson et al., 2019) and *OpenI* (Demner-Fushman et al., 2016), partitioned into 70% training and 30% testing. The study investigated two paradigms: Supervised Fine-tuning (SFT) (Ouyang et al., 2022) and a hybrid approach combining SFT with Direct Preference Optimization (DPO) (Rafailov et al., 2023). We examined following configurations:

- **SFT-50:** Trained using 50% of the available data to assess performance with limited supervision.
- **SFT-70:** Trained with 70% of the data to evaluate improvements with additional annotated data.
- **SFT-80:** Trained on 80% of the data to analyze performance with a higher level of supervision.
- **RGRO-50:** Applied SFT on 50% of the data, followed by DPO on the remaining 50%, to study the impact of preference-based alignment.
- **RGRO-70:** Combined 80% SFT with 20% DPO to examine the influence of a predominantly supervised approach with some preference optimization.
- **RGRO-80:** Utilized 70% SFT and 30% DPO to explore the effects of integrating preference optimization with substantial supervision.

**Experiment Objective:** The aim was to evaluate how each training strategy affects the model's ability to generate impressions that are accurate and better aligned with expert radiologists' expectations. By exploring different ratios of SFT and DPO, we sought to identify the optimal training methodology that balances accuracy and alignment with clinical preferences, enhancing the model's capability to produce high-quality radiological impressions.

**Experimental Results:**

Table 1: Performance comparison of models across two datasets using various metrics.

|  | OPENI | | | | | | MIMIC | | | | | |
|---|---|---|---|---|---|---|---|---|---|---|---|---|
|  | R1 | R2 | RL | BS-P | BS-R | BS-F1 | R1 | R2 | RL | BS-P | BS-R | BS-F1 |
| SFT-50 | 38.299 | 29.673 | 40.869 | 73.357 | 79.589 | 75.185 | 41.683 | 40.204 | 46.752 | 75.230 | 81.348 | 79.672 |
| SFT-70 | 46.721 | 28.996 | 45.263 | 81.146 | 86.849 | 83.619 | 53.159 | 48.106 | 52.592 | 82.492 | 87.805 | 85.093 |
| SFT-80 | 57.329 | 47.396 | 53.673 | 86.439 | 89.328 | 88.928 | 58.385 | 53.992 | 59.472 | 85.590 | 89.945 | 87.391 |
| RGRO-50 | 56.367 | 45.280 | 50.381 | 80.203 | 81.439 | 79.715 | 52.647 | 48.396 | 51.942 | 81.730 | 87.228 | 53.370 |
| RGRO-70 | 57.258 | 48.492 | 53.540 | 87.842 | 86.873 | 89.226 | 58.579 | 54.901 | 60.245 | 84.682 | 88.472 | 88.361 |
| RGRO-80 | 61.367 | 61.082 | 66.652 | 88.301 | 90.662 | 89.358 | 68.376 | 56.472 | 66.936 | 88.739 | 90.267 | 89.240 |

In Table 1, we present the performance comparison between models trained using supervised fine-tuning (SFT) and our proposed Radiology-Guided Reinforcement Optimization (RGRO) framework across two datasets: OPENI and MIMIC. The metrics used for evaluation include ROUGE (R1, R2, RL) and BERTScore (BS-P, BS-R, BS-F1).

For both datasets, we observe a consistent improvement in performance as we increase the proportion of training data used in the SFT models. Specifically, **SFT-80**, which utilizes 80% of the training data, significantly outperforms **SFT-50** and **SFT-70** in both ROUGE and BERTScore metrics. This trend suggests that larger training sets improve the model's ability to generate clinically accurate radiology impressions.

However, the introduction of RGRO, which integrates reinforcement learning into the model training process, further enhances the model's performance. For instance, **RGRO-80** achieves the highest scores across all metrics, particularly in BERTScore F1, where it attains values of **89.358** and **89.240** on OPENI and MIMIC, respectively. This demonstrates that by framing the generation of radiology

impressions as a sequential decision-making process, RGRO better aligns the model's outputs with clinical standards, outperforming models that rely solely on supervised fine-tuning.

The increase in both ROUGE and BERTScore metrics for RGRO-80 compared to SFT-80 highlights the effectiveness of incorporating reinforcement learning with expert-driven feedback in the training process. While supervised fine-tuning is able to capture language fluency, RGRO ensures that the generated outputs maintain clinical relevance, enhancing diagnostic precision.

## 5.3 PHASE 2: ZERO SHOT PERFORMANCE IN MULTI-CENTER GENERALIZATION

**Experimental Design:** In this phase, we tested the generalization ability of our model across multiple regions and clinical systems to evaluate how well it handles data from different clinical centers, including both domestic and international datasets. We conducted three sets of experiments:

- **Data Split 1:** The model was trained on a combination of private data from XiangYa Hospital (Zhong et al., 2023) and public datasets (MIMIC and OpenI). The test set was divided into three parts: MIMIC, OpenI, and the XiangYa dataset.

- **Data Split 2:** The model was trained on both the Xiangya private dataset and the combined MIMIC + OpenI datasets. Testing was performed separately on MIMIC, OpenI, and the Xiangya dataset, with the results stored in corresponding directories for each dataset.

**Experiment Objective:** The goal was to assess the model's ability to generate accurate and clinically relevant impressions across different clinical datasets, including its zero-shot capabilities. In addition to using automatic metrics from previous phases (Rouge, BLEU), we evaluated the model's performance on unseen datasets to test its zero-shot ability. Furthermore, we conducted a manual evaluation by having four radiologists review 500 samples each from the test datasets, providing additional insights into the clinical relevance and correctness of the model's output.

**Experimental Results:**

Table 2: Zero-Shot Performance trained on XiangYa Hospital Dataset

| | OPENI | | | | | | MIMIC | | | | | |
|---|---|---|---|---|---|---|---|---|---|---|---|---|
| | R1 | R2 | RL | BS-P | BS-R | BS-F1 | R1 | R2 | RL | BS-P | BS-R | BS-F1 |
| SFT-50 | 21.483 | 15.847 | 20.371 | 50.375 | 55.749 | 46.384 | 20.792 | 18.394 | 23.489 | 48.492 | 50.284 | 43.681 |
| SFT-70 | 34.284 | 21.583 | 28.384 | 62.589 | 60.385 | 52.478 | 40.738 | 32.582 | 30.472 | 65.396 | 69.372 | 58.385 |
| SFT-80 | 44.384 | 29.348 | 38.280 | 76.227 | 69.352 | 66.482 | 52.374 | 40.275 | 35.382 | 71.367 | 75.273 | 71.385 |
| RGRO-50 | 40.296 | 23.458 | 35.286 | 71.348 | 64.329 | 62.367 | 43.175 | 35.356 | 31.351 | 69.260 | 71.014 | 69.175 |
| RGRO-70 | 45.703 | 28.473 | 39.270 | 75.632 | 70.264 | 65.389 | 51.473 | 39.995 | 34.671 | 72.482 | 75.832 | 72.492 |
| RGRO-80 | 50.346 | 29.403 | 43.471 | 80.143 | 75.672 | 72.431 | 59.250 | 43.762 | 38.371 | 75.386 | 80.237 | 76.381 |

Table 2 demonstrates the zero-shot performance of models trained on the XiangYa Hospital dataset when evaluated on two external datasets: OPENI and MIMIC. This evaluation assesses the models' ability to generalize without being directly fine-tuned on the test datasets.

Our results reveal that while all models exhibit lower performance in the zero-shot setting compared to the supervised fine-tuning setup, the models trained with RGRO (RGRO-50, RGRO-70, RGRO-80) demonstrate stronger generalization capabilities than the SFT models. Notably, **RGRO-80** outperforms all other configurations in both ROUGE and BERTScore metrics on both datasets, achieving an **R1 score of 50.346** on OPENI and **59.250** on MIMIC. The superior performance of RGRO models in zero-shot settings suggests that reinforcement learning contributes to a more robust understanding of domain-specific language, allowing the model to generate clinically relevant impressions even when encountering previously unseen data.

This improvement highlights the advantage of aligning the model's decision-making process with domain-specific requirements, which is crucial for medical applications where accuracy and reliability are paramount.

## 5.4 QUALITATIVE ANALYSIS

The figures (Figure 2 and Figures 3 -6 in the Appendix) illustrate how the model processes and interprets radiological data across different anatomical contexts. The evaluation results indicate that

while our model shows slight improvements across both ROUGE and BERTScore metrics, the overall gains are modest. Specifically, the refined training strategy has contributed to better alignment with expert annotations, particularly in terms of semantic relevance as captured by BERTScore. This alignment reflects the model's enhanced ability to understand the nuances of clinical language and terminology. For example, the findings from the liver and lung systems highlight the model's capacity to generate clinically pertinent impressions based on the radiological data provided.

However, the improvements observed do not represent a significant leap in performance compared to earlier evaluations. The current results suggest that, although the model's ability to generate clinically relevant impressions has been enhanced, further optimization may be required to achieve more substantial advancements. This could involve exploring additional training datasets, refining the model architecture, or incorporating more sophisticated reinforcement learning techniques to better capture the intricacies of clinical reasoning.

Additionally, the qualitative insights from this analysis emphasize the importance of continuous feedback from medical professionals. Regular evaluations in diverse clinical scenarios are essential to ensure that the model remains robust and reliable in real-world applications. Such iterative adjustments will be vital for enhancing the model's performance and ensuring it effectively supports clinical decision-making. Overall, while the current results are promising, they also underscore the need for ongoing research and development to fully leverage the potential of machine learning in the clinical domain.

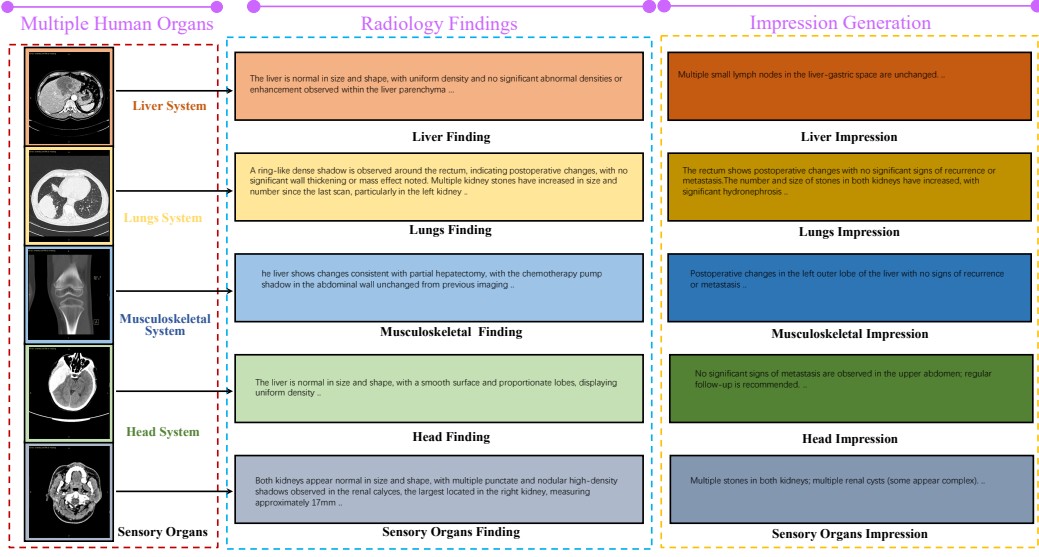

Figure 2: **The First Visualization of Radiology Finding and Impression.** We present five body-systems impressions generated by RGRO coupled with radiology findings. Each row shows the detailed process of each stage of the different diagnosis on systems of liver, lung, musculoskeletal system, head and sensory organs.

## 6 DISCUSSION & CONCLUSION

In this paper, we presented Radiology-Guided Reinforcement Optimization (RGRO), a fundamental reinforcement learning-based framework designed to optimize large language models for radiology reports. By framing the generation process as a sequential decision-making task, RGRO enables the model to better align with clinical requirements, moving beyond traditional likelihood-based objectives. Our empirical results demonstrate that RGRO improves diagnostic accuracy and clinical utility compared to supervised fine-tuning and other baseline models.

While RGRO shows significant promise in enhancing the alignment of LLMs with radiology-specific objectives, several limitations must be acknowledged. First, the scope of our experiments was limited to a subset of radiology tasks, primarily focusing on diagnostic impressions derived

from radiology reports. Future work could explore more diverse medical specialties and imaging modalities to further validate the generalizability of our approach. Additionally, our reliance on expert-labeled datasets, though partially mitigated by reinforcement learning, still presents challenges in terms of scalability, as large-scale expert annotations remain costly and time-consuming.

Another limitation lies in the potential problem of overoptimization, a common issue in reinforcement learning when models become excessively tuned to specific feedback, diverging from the broader distribution of acceptable outputs. In this work, we did not explicitly address this concern, leaving it as an important direction for future research. Expanding the framework to include mechanisms for preventing overfitting to specific datasets or feedback loops is critical for ensuring robustness across diverse clinical environments. Future iterations could incorporate more dynamic and interactive feedback mechanisms, enabling models to adapt continuously to evolving clinical guidelines and practices.

RGRO is cost-effective compared to traditional approaches. Moreover, the use of reinforcement learning in specialized domains provides a flexible mechanism to incorporate real-time feedback and evolving clinical standards. While the RGRO framework shows great potential in generating better radiology reports through reinforcement learning, further exploration into data diversity, feedback mechanisms, and the prevention of overoptimization is necessary. We hope this work provides a foundational step toward targeted optimization of LLMs for medical applications and improves the practical utility of AI-driven radiology solutions.

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

## A APPENDIX

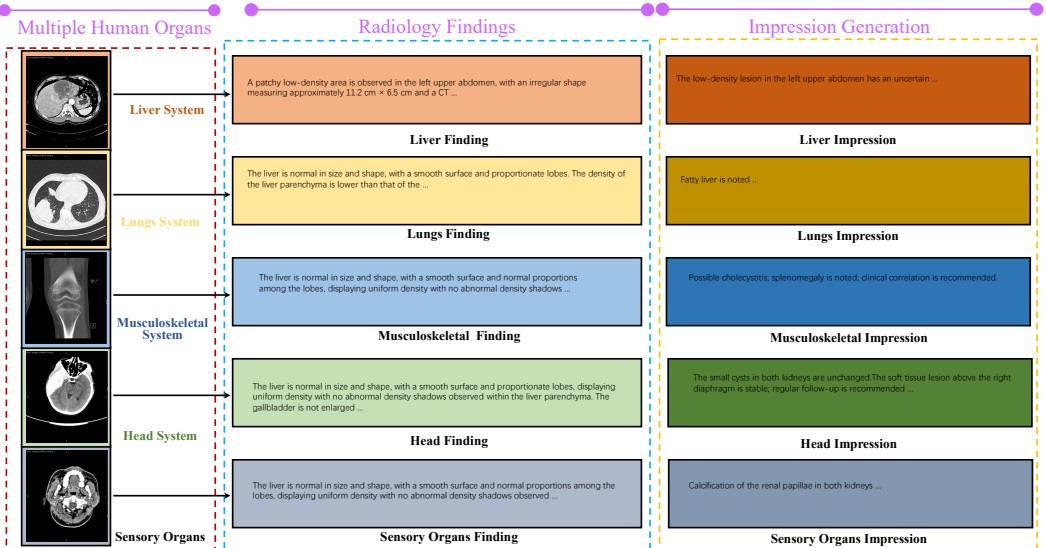

Figure 3: **The Second Visualization of Radiology Finding and Impression.** We present five body-systems impressions generated by RGRO coupled with radiology findings. Each row shows the detailed process of each stage of the different diagnosis on systems of liver, lung, musculoskeletal system, head and sensory organs.

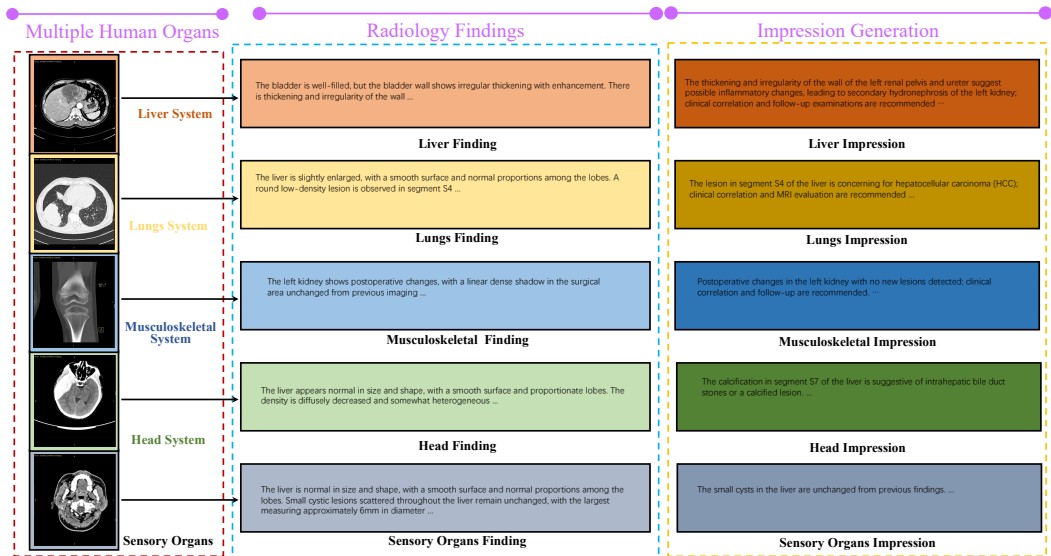

Figure 4: **The Third Visualization of Radiology Finding and Impression.** We present five body-systems impressions generated by RGRO coupled with radiology findings. Each row shows the detailed process of each stage of the different diagnosis on systems of liver, lung, musculoskeletal system, head and sensory organs.

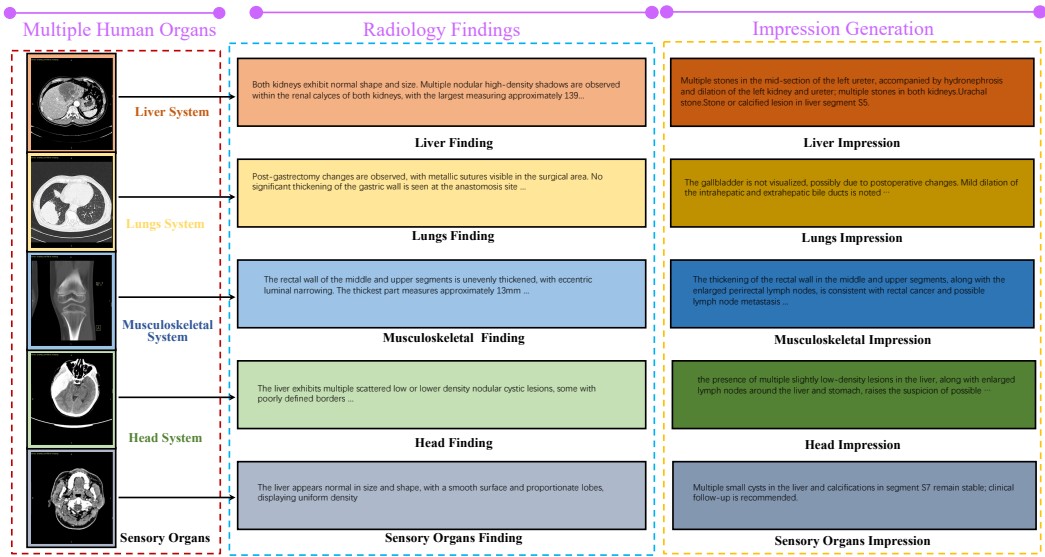

Figure 5: **The Forth Visualization of Radiology Finding and Impression.** We present five body-systems impressions generated by RGRO coupled with radiology findings. Each row shows the detailed process of each stage of the different diagnosis on systems of liver, lung, musculoskeletal system, head and sensory organs.

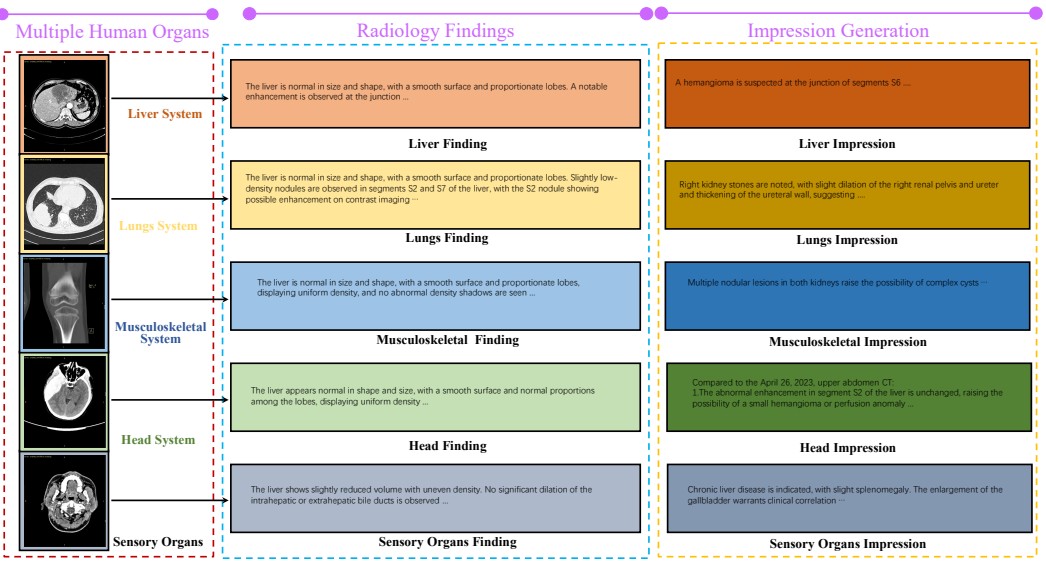

Figure 6: **The Fifth Visualization of Radiology Finding and Impression.** We present five body-systems impressions generated by RGRO coupled with radiology findings. Each row shows the detailed process of each stage of the different diagnosis on systems of liver, lung, musculoskeletal system, head and sensory organs.

