# OpenReview forum: "An Efficient LLM Alignment Framework for Automated Radiology Impression Generation"
_ICLR.cc/2025/Conference — Submitted to ICLR 2025_

### Official Review · Reviewer_95oJ · 2024-10-27

**Soundness:** 2
**Presentation:** 2
**Contribution:** 2
**Rating:** 3
**Confidence:** 4

**Summary:**

The study introduced radiology-guided reinforcement optimization (RGRO), a framework to finetune LLMs and align with human preference for radiology report generation (generating the impression section from the findings section). The framework utilizes the instruction fin-tuning with low-rank adaptation (LoRA), then reinforcement learning with direct preference optimization (DPO) guided by an LLM parser to align the model outputs with human expert preferences. The authors evaluated RGRO on MIMIC and OpenI, and compared with SFT baselines, and also evaluated the model’s zero-shot performance on a private dataset. The authors claimed that RGRO significantly enhances the diagnostic precision and clinical utility of reports generated by LLMs, and outperforms SFT and other SoTA models.

**Strengths:**

* The authors proposed RGRO, adopting LoRA and DPO, and is specifically tailored for radiology report generation using radiology relevant objectives in its customized reward function. This may help align LLMs with domain specific needs beyond stylish alignment.
* With the LLM parser and DPO, RGRO may mitigate the requirement of collecting large annotated datasets.
* Ablation of understanding the SFT and DPO setting is good.

**Weaknesses:**

* No comparison with other SoTA models even though the authors mentioned that RGRO outperforms state-of-the-art models. It would be nice to have the results of comparison listed in the main context.
* More details of the evaluation dataset setup might be needed. The author mentioned the phase 1 experiment they use a radiology dataset to do pretraining, finetuned using MIMIC and OpenI, then evaluated on the MIMIC and OpenI test split. In phase 2 the model was trained in two ways (but it seems the same?) using XiangYa + MIMIC + OpenO datasets, and evaluated on the XiangYa test split. It is better to make the description clearer, e.g. which dataset you used for phase 1 pretraining, the number of examples in each split / each dataset, why you used different data split for training for each stage, what’s the benefit of experimenting two different sets of experiment? This would be helpful to understand the generalizability of the proposed method.
* The ablation of modified DPO versus standard DPO might also be helpful for readers to understand how much the modification can help.
* Details of the LLM parser, such as the instruction prompts used for preference selection, is important for reproducibility.
* The authors used ROUGE and BERTScore for evaluation, which are helpful NLG metrics yet may not fully capture the clinical utility. Using clinically relevant metrics, or having radiologist to do evaluation would be able to strengthen the authors’ claim of better diagnostic precision and clinical utility.  For example, in this paper https://openaccess.thecvf.com//content/CVPR2023/papers/Tanida_Interactive_and_Explainable_Region-Guided_Radiology_Report_Generation_CVPR_2023_paper.pdf table 1 and 2, also https://arxiv.org/pdf/2311.18260 table 1, these are commonly used NLG metrics and also clinical relevant metrics such as RadGraph F1, from different evaluation perspectives.

**Questions:**

* RL is known to have the risk of overoptimization, how do the authors monitor and control this risk? I.e., when to stop the DPO?
* How was the LLM parser’s performance evaluated in terms of agreement with human preferences?
* It would be nice to discuss how “RGRO focuses on enhancing fine-grained reasoning” via the modified DPO process.
* It seems to me that section 3.1 to 3.3 is a general RL introduction, perhaps it’s better to move them into the appendix if you prefer to keep them, and add more other details (e.g., datasets) into the main context. You may also move the LoRA introduction in 4.1 to the appendix.
* Given that the authors used GPT for evaluation, will the evaluation be stable while changing the GPT version? (i.e., always accept / reject the expert annotated good / bad impression)
* For MIMIC, are you using MIMIC-CXR or MIMIC-IV CXR images?
* Please fix: I suppose that your description of RGRO-70 and RGRO-80 are reversed? (RGRO-70 should be 70%SFT + 20%DPO I believe)
* Some numbers are pretty close, do authors ensure that the results are statistically significant? E.g. did authors use bootstrapping to compute confidence intervals?
* Did authors see any specific cases that RGRO performed very well / very bad?
* Did authors do a hyperparameter search for DPO? E.g. Beta value in DPO reward function.
* How sensitive is RGRO to the choice of hyperparameters (e.g., β in the DPO loss function)? Did you perform a hyperparameter search, and if so, what were the results?

**Details Of Ethics Concerns:**

No ethics concerns

---

> ### Author Response · Authors · 2024-11-22
>
> We thank the reviewer for insightful comments. We have carefully considered your comments and responded to the individual concerns.
>
> –RL is known to have the risk of over optimization, how do the authors monitor and control this risk? I.e., when to stop the DPO?
>
> This is a good point. Because of the limit of time, we did not design that detailed experiment, and in the future we plan to check through to compare the win rate between the DPO output and the fine-tuning output. to see whether the DPO model has been overoptimization. Also, increasing the diversity of the training samples could help this situation.
>
> –How was the LLM parser’s performance evaluated in terms of agreement with human preferences?
> We design the specific prompt to run for the preference comparison. The prompt was designed with the involvement of a real doctor. Due to time constraints, we were unable to include those samples in the paper, but we will incorporate them in a future version.
>
>
>
> –It would be nice to discuss how “RGRO focuses on enhancing fine-grained reasoning” via the modified DPO process.
> Thank you for the suggestion. Since the framework we propose involves fine-tuning first and then performing DPO, we believe this approach is more effective than simply modifying the DPO process. We will make adjustments in a future version.
>
>
>
> –It seems to me that section 3.1 to 3.3 is a general RL introduction, perhaps it’s better to move them into the appendix if you prefer to keep them, and add more other details (e.g., datasets) into the main context. You may also move the LoRA introduction in 4.1 to the appendix.
> Thank you for the suggestion. We will modify it in the future version.
>
>
>
> –Given that the authors used GPT for evaluation, will the evaluation be stable while changing the GPT version? (i.e., always accept / reject the expert annotated good / bad impression)
> Thank you for the suggestion. We will add the ablation study on these in the future version.
>
>
> –For MIMIC, are you using MIMIC-CXR or MIMIC-IV CXR images?
> We did not use an image dataset; we only used a text dataset. Thank you for your suggestion.
>
>
> –Please fix: I suppose that your description of RGRO-70 and RGRO-80 are reversed? (RGRO-70 should be 70%SFT + 20%DPO I believe)
> Thank you for the suggestion. We will modify it in the future version.
>
>
> –Some numbers are pretty close, do authors ensure that the results are statistically significant? E.g. Did authors use bootstrapping to compute confidence intervals?
> Your suggestion is excellent, but due to time and computational resource constraints, we were unable to include that part of the experiment.
>
>
>
> –Did authors see any specific cases that RGRO performed very well / very bad?
> Thank you for suggesting we will add the specific case study session in the future version.
>
>
> –Did authors do a hyperparameter search for DPO? E.g. Beta value in DPO reward function.
> Thank you for your suggestion. We pick the best parameter following the DPO paper.
>
>
> –How sensitive is RGRO to the choice of hyperparameters (e.g., β in the DPO loss function)? Did you perform a hyperparameter search, and if so, what were the results?
> Thank you for your suggestion. We did not do that experiment since the training time is long. We will add them in the future version.

---

> > ### Comment · Reviewer_95oJ · 2024-11-25
> >
> > Thank you authors for the explanations. I would prefer to keep the original rating at the moment since most of the issues and questions may need to be addressed in the next version.

---

### Official Review · Reviewer_kkEW · 2024-11-03

**Soundness:** 2
**Presentation:** 1
**Contribution:** 1
**Rating:** 3
**Confidence:** 4

**Summary:**

The manuscript presents a framework called RGRO for optimizing large language models in generating radiology impressions from findings. The framework consists of two phases: instruction fine-tuning with LoRA and reinforcement learning with DPO, guided by feedback from an LLM parser. The authors evaluated their approach on datasets such as MIMIC and OPEN-I, and demonstrated a comparison between RGRO and SFT.

**Strengths:**

1. Introducing a reinforcement learning method into the report generation task is a reasonable approach to addressing the limitations of next-token prediction tasks in large language models (LLMs) for clinical diagnosis generation.
2. Experiments were conducted on different datasets and in a multi-center setting, providing a robust evaluation.

**Weaknesses:**

1. Problem Definition: In radiology, findings and impressions describe the image from both an observational and diagnostic perspective. I strongly question the significance and feasibility of generating impressions solely from text findings without considering the image. For example, opacities in a chest X-ray may lead to different preferred diagnoses depending on the image feature, clinical indication, and patient context.
2. Framework Confusion: The framework is somewhat unclear. In the PRELIMINARIES and RADIOLOGY-GUIDED REINFORCEMENT OPTIMIZATION sections, PPO and DPO are mentioned alternately, and the specific method used by RGRO is not clearly defined. Furthermore, the PRELIMINARIES section does not even mention the DPO method.
3. Lack of Novelty: The method lacks innovation, as it appears to be a direct application of DPO to a paired dataset.
4. Lack of Implementation Details: There is insufficient detail regarding the dataset and preprocessing steps, the construction and validation of the LLM parser, and the experimental hyperparameters and training methods.
5. Insufficient Experimental Evaluation: The experimental evaluation is lacking, with no comparison against other methods. The design and analysis of the two experimental groups are also flawed.

**Questions:**

1. As mentioned, is generating impressions solely from text-based findings feasible and useful?
2. What specific method is used during the reinforcement learning phase? This should be clarified.
3. If DPO is used in the second phase, were any modifications or optimizations applied? How does RGRO compare to other DPO frameworks?
4. What are the implementation details of the LLM parser? Specifically, which LLM was used, and how were the prompts designed, aligned, and evaluated?
5. In the experiments involving RGRO, what hyperparameters were used and how were they determined? Additionally, how were checkpoints selected, and what criteria were used? These should are be clarified.
6. In section 5.2, the comparison between the SFT series and the RGRO series seems unfair. For example, SFT-80 uses only 80% of the data, while RGRO-80 uses the entire dataset, with 80% for SFT and 20% for DPO.
7. In section 5.2, RGRO-80 appears to outperform RGRO-70 and RGRO-50. However, according to the manuscript, RGRO-80 consists of 80% SFT and 20% DPO. Does this indicate a trend where less DPO leads to better performance?
8. In section 5.3, the two data splits seem to be set up similarly. What are the differences between these settings?


Minor

1. In line 348, RGRO-80: Utilized 70% SFT and 30% DPO—is this a typo?
2. In Figure 2, under Musculoskeletal Finding: "The liver shows changes consistent with partial hepatectomy, with the chemotherapy pump shadow in the abdominal wall unchanged from previous imaging..."—this is not a musculoskeletal finding.

---

> ### Author Response · Authors · 2024-11-22
>
> We thank the reviewer for insightful comments. We have carefully considered your comments and responded to the individual concerns.
>
> –As mentioned, is generating impressions solely from text-based findings feasible and useful?
> Generating impressions from text-based findings is feasible and can be useful in assisting radiologists by summarizing detailed findings into concise diagnostic impressions. Additionally, you are correct there has the image to text dataset but we did not use that in this work. Because our work is focused on the text part so we did not include the experiment with the image input. We will add them in the future version if the computing resource  is enough. Thank you for the suggestion.
>
> –What specific method is used during the reinforcement learning phase? This should be clarified.
> We used Direct Preference Optimization (DPO) during the reinforcement learning phase of RGRO. We will clarify this in the methodology section and provide a detailed explanation of how DPO was implemented in our framework.
> We adapted the standard DPO approach by incorporating radiology-specific reward functions guided by expert feedback via the LLM parser.
>
>
> –If DPO is used in the second phase, were any modifications or optimizations applied? How does RGRO compare to other DPO frameworks?
> In the tradition DPO needs preference for dataset labels by humans, but we label the dataset using the LLM parser. RGRO is proposed that should do the fine-tuning then doing the reinforcement alignment.
>
>
> –What are the implementation details of the LLM parser? Specifically, which LLM was used, and how were the prompts designed, aligned, and evaluated?
> We are using the GPT-4o-mini as backbone for the LLM parser. These prompts are designed with the help of the professional doctor to help do a profession preference dataset. Our prompt design strictly adheres to the prompt technique; however, we have not conducted experiments to explore different prompt techniques. Currently, we use the raw prompts designed by real doctors. For evaluation, we focus on identifying the preferred outputs generated by the DPO.
>
>
>
> –In the experiments involving RGRO, what hyperparameters were used and how were they determined? Additionally, how were checkpoints selected, and what criteria were used? These should be clarified.
>
> We apologize for the confusion. Due to time constraints, we were unable to design that specific experiment. However, we plan to conduct it in the future. Thank you for your advice.
>
>
>
>
>
> –In section 5.2, the comparison between the SFT series and the RGRO series seems unfair. For example, SFT-80 uses only 80% of the data, while RGRO-80 uses the entire dataset, with 80% for SFT and 20% for DPO.
> That is a good point. However, for the DPO part, we only need to utilize the findings, not the impression. The impression can be generated by the previously fine-tuned model, and a LLM parser can be used to determine the preference.
>
>
>
> –In section 5.2, RGRO-80 appears to outperform RGRO-70 and RGRO-50. However, according to the manuscript, RGRO-80 consists of 80% SFT and 20% DPO. Does this indicate a trend where less DPO leads to better performance?
> Sorry for the confusion, because the limit of time to do the experiment, we could get the conclusion for that. In the future, if we do more experiments, we would discuss that.
>
>
> –In section 5.3, the two data splits seem to be set up similarly. What are the differences between these settings?
>
> Sorry for the confusion. That part was the writing problem we will modify in the future version.

---

> > ### Comment · Reviewer_kkEW · 2024-11-26
> >
> > Thank you for your response. However, several issues remain unclear, even without considering the value of generating impressions solely from findings text. The authors have not explained the specific experimental settings like hyperparameters, method for selecting checkpoints in experiment section. And the manuscript contains several ambiguous unclear and not responded.  e.g.  in Section 5.3, line 389, "We conducted three sets of experiments," is followed by only two data splits (and which seems to be very similar?).  Additionally, the fairness issue in the comparison between the SFT and DPO groups in Section 5.2 has not been adequately addressed. These require substantive responses, not merely writing problem.
> > Given these issues, I am inclined to maintain my original opinion.

---

### Official Review · Reviewer_mJYg · 2024-11-03

**Soundness:** 3
**Presentation:** 3
**Contribution:** 3
**Rating:** 5
**Confidence:** 3

**Summary:**

This paper explores enhancing radiology LLMs through Radiology-Guided Reinforcement Optimization (RGRO), a framework that replaces traditional supervised fine-tuning with a reinforcement learning approach for RRG. RGRO treats report generation as a series of decision-making steps, optimizing for radiology-specific objectives such as diagnostic consistency and adherence to clinical guidelines. Experimental results show that RGRO improves report quality over existing methods.

**Strengths:**

The paper is well-written and relatively easy to follow with the figures. Problem was established and context was provided nicely so the need for RGRO is clear.

**Weaknesses:**

Small things:
- "Fin-tuning" spelt wrong in Figure 1, A.1.
- Table 1 should highlight the best performing cells to make it easier to read

Evaluation with other metrics (aside from just ROUGE and BERTScore) would be desirable as they are not the most suitable for clinical text and there are several other established factuality metrics for radiology and also LLM-based metrics. I recommend the authors do experiments with GREEN (https://arxiv.org/html/2405.03595v1), FineRadScore (https://arxiv.org/html/2405.20613v2), and G-Rad (https://arxiv.org/html/2403.08002v2), as these are relevant.

Perhaps some equations of existing methodolgies are unneeded as they don't add to the overall message of the paper. However, it doesn't really harm the paper - if more space is needed for further experiments or validation, I'd remove some general explanations in Preliminaries and focus more on the modified policy for RGRO.

**Questions:**

Reccomend authors do further validation on their results by using factuality metrics or LLM-based metrics.

---

> ### Author Response · Authors · 2024-11-22
>
> We thank the reviewer for insightful comments. We have carefully considered your comments and responded to the individual concerns.
>
> –Recommend authors do further validation on their results by using factuality metrics or LLM-based metrics.
> We appreciate the recommendation to conduct further validation using factuality metrics and LLM-based metrics. In the revised manuscript, we will include experiments utilizing experiments like the factuality metrics[1].
>
>
>
> [1] Zha, Y., Yang, Y., Li, R., Hu, Z. AlignScore: Evaluating Factual Consistency with a Unified Alignment Function. arXiv preprint arXiv:2305.16739, ACL 2023.

---

> > ### Comment · Reviewer_mJYg · 2024-11-23
> >
> > I would like to see the results of said experiments, instead of just "we will do them...". Authors also did not acknowledge any LLM-based metrics.

---

> > > ### Author Response · Authors · 2024-11-24
> > >
> > > We thank the reviewer for their insightful comments. We sincerely apologize for not including the additional experiment. This was due to issues with our computing resources, which prevented most of our experiments from proceeding.

---

### Official Review · Reviewer_VzfC · 2024-11-05

**Soundness:** 3
**Presentation:** 2
**Contribution:** 2
**Rating:** 5
**Confidence:** 4

**Summary:**

The paper presents Radiology-Guided Reinforcement Optimization (RGRO), a framework that uses reinforcement learning to align large language models with the specific requirements of radiological practice, optimizing directly for radiology-focused objectives.
Empirical results show that RGRO enhances diagnostic precision and clinical utility in radiology reports, compared to traditional supervised methods while incorporating expert feedback and diagnostic tools without large annotated datasets.

**Strengths:**

- The paper proposes Radiology-Guided Reinforcement Optimization (RGRO), a tailored policy optimization framework designed specifically for radiology.
- The papers utilizes three datasets in the experimentation.
- RGRO- 80 outperforms all other configurations in both ROUGE and BERTScore.

**Weaknesses:**

- There are several new methods that could be tested in addition to DPO and PPO in the ablation experiments.
- Few-shot learning experiments could be tested to comparer the performance with the SFT model etc.in addition to the zero-shot methods.
- The results section only compares SFT with the proposed method. Comparing the papers proposed method with the current DPO and PPO methods could be help with comparison.
- Why is the data trained on the Hospital Data and tested on OpenI and MIMIC only, other variations/data could be trained on?
- The diagram has an error 'Fin-tuning'-->'Fine-tuning'.
- The paper could be written and structured better.

**Questions:**

- It is not clear what the 50,70, and 80 refer too? Why does the 70 have additional annotated data.
- Where clinicians involved in expert incorporation and what particular aspects of expert feedback incorporation was added, such as the format of the feedback.

---

> ### Author Response · Authors · 2024-11-22
>
> We thank the reviewer for insightful comments. We have carefully considered your comments and responded to the individual concerns.
>
> –It is not clear what the 50,70, and 80 refer to? Why does the 70 have additional annotated data?
> The numbers 50, 70, and 80 refer to the percentages of the dataset used during the Supervised Fine-Tuning (SFT) phase in our experiments. Specifically:
> RGRO-50: 50% of the data was used for SFT, and the remaining 50% was used for the DPO phase.
> RGRO-70: 70% of the data was used for SFT, and 30% for DPO.
> RGRO-80: 80% of the data was used for SFT, and 20% for DPO
> The 70% did not contain additional annotated data. Sorry for the confusion, we will pay more attention in the future version.
>
> –Where clinicians were involved in expert incorporation and what particular aspects of expert feedback incorporation was added, such as the format of the feedback.
> A portion of our dataset includes handwritten feedback directly from hospital clinicians. These handwritten notes provided rich, contextually relevant insights that were instrumental in training and evaluating our model. And for the DPO part doctors have joined to help align with the RLHF(Reinforcement Learning From Human Feedback) architecture.

---

### Comment · Area_Chair_LyQv · 2024-11-23
**Please review author response**

Dear reviewer,

Could you review the author response and let them know if you are satisfied with it or if you have any additional questions?

Kind regards,

Your AC

---

### Meta-Review · Area_Chair_LyQv · 2024-12-20

**Metareview:**

This work uses reinforcement learning to improve the alignment of LLM for medical language tasks. In particular, it proposes a policy optimisation framework called radiology-guided reinforcement learning to automatically generate the impression part from the finding part of a radiology report. Experimental study is conducted on several datasets to show the efficacy of the proposed method. Reviewers comment that this work is well written, presents the motivation clearly, is a reasonable approach, and does a good ablation study. At the same time, reviewers raise issues related to the limited experimental comparison, the need of using more suitable metrics, the soundness of generating impressions from findings, the lack of novelty, and the need of more implementation details, etc. The authors provide a rebuttal. However, the rebuttal is not convincing enough and does not fully address the raised issues, which is also reflected in the reviewers’ comments after reading the rebuttal. All the final ratings are on the negative side. By checking the submission, the reviews, and the rebuttals, AC agrees with the reviewers on the raised concerns. Therefore, this work in its current form cannot be recommended for acceptance.

**Additional Comments On Reviewer Discussion:**

Reviewers raise issues related to the limited experimental comparison, the need of using more suitable metrics, the soundness of generating impressions from findings, the lack of novelty, and the need of more implementation details, etc. The authors provide a rebuttal. However, the rebuttal is not convincing enough and does not fully address the raised issues, which is also reflected in the reviewers’ comments after reading the rebuttal. By checking the submission, the reviews, and the rebuttals, AC agrees with the reviewers on the raised concerns and that the responses in the rebuttal are not sufficient.

---

### Decision · Program_Chairs · 2025-01-22

Reject